# Differential in-hospital mortality and intensive care treatment over time: Informing hospital pathways for modelling COVID-19 in South Africa

Lise Jamieson[1,2,3]*, Cari Van Schalkwyk[3], Brooke E. Nichols[1,2,4,5], Gesine Meyer-Rath[1,3,4], Sheetal Silal[6,7], Juliet Pulliam[3], Lucille Blumberg[8,9], Cheryl Cohen[8,10], Harry Moultrie[8], Waasila Jassat[8,9]

1 Health Economics and Epidemiology Research Office (HE2RO), Department of Internal Medicine, Faculty of Health Sciences, University of the Witwatersrand, Johannesburg, South Africa, 2 Department of Medical Microbiology, Amsterdam University Medical Center, Amsterdam, The Netherlands, 3 The South African Department of Science and Innovation/National Research Foundation Centre of Excellence in Epidemiological Modelling and Analysis (SACEMA), Stellenbosch University, Stellenbosch, Republic of South Africa, 4 Department of Global Health, School of Public Health, Boston University, Boston, Massachusetts, United States of America, 5 Foundation for Innovative New Diagnostics, Geneva, Switzerland, 6 Modelling and Simulation Hub, Africa, Department of Statistical Sciences, University of Cape Town, Cape Town, South Africa, 7 Centre for Tropical Medicine and Global Health, Nuffield Department of Medicine, University of Oxford, Oxford, United Kingdom, 8 National Institute for Communicable Diseases, National Health Laboratory Service, Johannesburg, South Africa, 9 Right to Care, Centurion, South Africa, 10 School of Public Health, Faculty of Health Sciences, University of the Witwatersrand, Johannesburg, South Africa

* ljamieson@heroza.org

**Data Availability Statement:** The deidentified individual participant dataset used in this article will

## Abstract

There are limited published data within sub-Saharan Africa describing hospital pathways of COVID-19 patients hospitalized. These data are crucial for the parameterisation of epidemiological and cost models, and for planning purposes for the region. We evaluated COVID-19 hospital admissions from the South African national hospital surveillance system (DATCOV) during the first three COVID-19 waves between May 2020 and August 2021. We describe probabilities and admission into intensive care units (ICU), mechanical ventilation, death, and lengths of stay (LOS) in non-ICU and ICU care in public and private sectors. A log-binomial model was used to quantify mortality risk, ICU treatment and mechanical ventilation between time periods, adjusting for age, sex, comorbidity, health sector and province. There were 342,700 COVID-19-related hospital admissions during the study period. Risk of ICU admission was 16% lower during wave periods (adjusted risk ratio (aRR) 0.84 [0.82–0.86]) compared to between-wave periods. Mechanical ventilation was more likely during a wave overall (aRR 1.18 [1.13–1.23]), but patterns between waves were inconsistent, while mortality risk in non-ICU and ICU were 39% (aRR 1.39 [1.35–1.43]) and 31% (aRR 1.31 [1.27–1.36]) higher during a wave, compared to between-wave periods, respectively. If patients had had the same probability of death during waves vs between-wave periods, we estimated approximately 24% [19%-30%] of deaths (19,600 [15,200–24,000]) would not have occurred over the study period. LOS differed by age, ward type (ICU stays were longer than non-ICU) and death/recovery outcome (time to death

be made available with publication upon reasonable request. Proposals should be directed to michelleg@nicd.ac.za. Data will be provided upon provision of protocol and ethics approval for the proposed study and a signed data sharing agreement.

**Funding:** The work of GMR and LJ on the SACMC has been made possible by the generous support of the American People and the President's Emergency Plan for AIDS Relief (PEPFAR) through the United States Agency for International Development (USAID) under the terms of Cooperative Agreement 72067419CA00004 to HE2RO. The contents are the responsibility of the authors and do not necessarily reflect the views of PEPFAR, USAID or the United States Government. https://www.state.gov/pepfar/. SPS is funded by the Wellcome Trust (GN: 2114236/Z/18Z) and the Clinton Health Access Initiative. CVS and JRCP are supported by the Department of Science and Innovation and the National Research Foundation. Any opinion, finding, and conclusion or recommendation expressed in this material is that of the authors, and the NRF does not accept any liability in this regard. The SACMC's work is also supported by the Bill & Melinda Gates Foundation under Investment INV-035464. The views and opinions expressed in this report do however not necessarily reflect the positions or policies of the Bill & Melinda Gates Foundation.

**Competing interests:** The authors have declared that no competing interests exist.

was shorter in non-ICU); however, LOS remained similar between time periods. Healthcare capacity constraints as inferred by wave period have a large impact on in-hospital mortality. It is crucial for modelling health systems strain and budgets to consider how input parameters related to hospitalisation change during and between waves, especially in settings with severely constrained resources.

## Introduction

The novel severe acute respiratory syndrome coronavirus 2 (SARS-CoV-2) was confirmed as the causative agent of coronavirus disease 2019 (COVID-19) in December 2019, which was declared a global pandemic by the World Health Organisation (WHO) on 11 March 2020 [1]. A number of countries imposed lockdown restrictions in order to stall the spread of the virus to prevent the surge of severe and critical cases from overwhelming the healthcare system. Since then COVID-19 has had devastating health and economic consequences globally [2]. South Africa reported its first case in March 2020 which was followed by a nationwide lockdown soon after [3]. The first wave of COVID-19 cases peaked in August 2020, followed by a second wave attributed to the more transmissible Beta [4] which peaked in January 2021 [5]. A third wave, attributed to the Delta variant, began around May 2021, peaking nationally in August 2021 [6]. During the waves, several restrictions were put in place in order to mitigate the spread of the virus [7]. The South African COVID-19 vaccination roll-out [8, 9], having started in February 2021 with priority group 1 –healthcare workers–had by the start of the third wave only reached around 320,000 individuals, almost exclusively in healthcare workers who had received a single dose of the Jansen vaccine, a small portion compared to the 43 million South Africans >15 years [10]. By the peak of the third wave in mid-August 2021, more priority groups were included in the vaccination roll-out, with approximately 7 million individuals having had at least one dose of the Pfizer vaccine, with 43% and 24% of these aged 50–59 and 60+ years, respectively [11].

The South African COVID-19 Modelling Consortium (SACMC), a group of infectious disease modellers and epidemiologists from academic, non-profit, and government institutions across South Africa, was established in late March 2020. Coordinated by the National Institute for Communicable Diseases (NICD), on behalf of the National Department of Health (NDOH), the SACMC developed the National COVID-19 Epi Model (NCEM) in March 2020 to project the spread of the disease and to support policy development, planning and budgeting in South Africa over the coming months [12]. The NCEM is a compartmental transmission model designed to model the COVID-19 pandemic; it follows a generalised Susceptible-Exposed-Infectious-Recovered (SEIR) structure accounting for disease severity (asymptomatic, mild, severe and critical cases) and different treatment pathways (outpatients, intensive care unit (ICU) and non-ICU care). The model aimed to inform resource requirements, including inpatient beds, staff and ventilation equipment, and predict where gaps could arise given limited available resources within the health system at the start of the epidemic. In the absence of sufficient data in the early stages of the pandemic, the NCEM was first parameterized using international data [13, 14], but re-parameterized with local data as soon as it became available [15]. Over the course of the epidemic in South Africa, the NCEM was adapted to account to account for differences in age distribution, geography, comorbidities, variants in circulation and vaccination status.

There have been limited published data from low- and middle-income countries (LMIC) describing COVID-19-related hospitalisation and inpatient care, especially beyond the first wave of COVID-19 in 2020. One study described a cohort of public sector COVID-19 patients in one of South Africa's nine provinces, Western Cape, as observed at the beginning of the first wave [16]. Though the study does not distinguish between ICU and non-ICU care, it found that patients aged 40 years or older had a 2- to 7-fold higher in-hospital mortality compared to younger patients (<40 years). Jassat et al [17] found a 3- to 8-fold increase in odds of mortality in older age groups, and a 67% increased risk for in-hospital mortality in patients with any comorbidity compared to patients with no known comorbidities. Overall, there was a 31% increased risk of in-hospital mortality in the second wave as compared to the first wave [17]. Solanki et al [18] report on the private sector experience in South Africa, and also found an increased risk of hospitalisation and mortality amongst those with comorbidities and older age groups, and a variable risk of mortality between provinces. A review of studies reporting lengths of stay across multiple countries found only a single study in Egypt, and it had comparatively shorter length of stay compared to other countries [19]. More recent studies focused on the fourth wave, dominated by the less severe Omicron variant [20–24].

As the COVID-19 pandemic progressed, the standard-of-care treatment as well as the healthcare-seeking behaviour of individuals changed in response to the limited available capacity. This analysis aims to describe hospital parameter inputs as they relate to the NCEM, within non-ICU and ICU care pathways, age groups, and public/private healthcare sector. Further, we compared these parameters estimated during the wave periods to those between waves in order to assess the impact waves could have played in mortality and admission to ICU care when the healthcare system was under strain.

## Methods

### Hospital surveillance data in South Africa

DATCOV, a national sentinel hospital surveillance dataset initiated in April 2020, records basic demographic information and treatment-specific data for all COVID-19 associated hospitalisations in South Africa from all private and public sector hospitals nationally [25]. We used data exported on 6 September 2021 to estimate the probability of ICU admission, mechanical ventilation, and death in non-ICU and ICU. Lengths of stay from admission to death or recovery were estimated separately for ICU and non-ICU admissions. In order to avoid bias towards those recently admitted or healthier patients with shorter lengths of stay and/or better outcomes, we right-censored the data to include patients who were admitted >30 days prior to the data export (i.e., up to 7 August 2021) to allow for sufficient time for patients to experience an outcome.

For each province, we classified time into periods relative to COVID-19 waves based on reported case data (Table A in S1 Text). Based on wave metrics established by the South African Ministerial Advisory Committee on COVID-19 [26] and a provincial-level analysis of reported case numbers, the wave start was defined as the start of a period of sustained increase prior to the declaration of the wave, and the wave end date set as the date at which the end-of-wave threshold was reached (or data export for provinces which were still in their third wave) (for more information on the definitions, see [27]). Starting from the first wave, patients were classified into time periods based on their admission date, into wave 1, post-wave 1, wave 2, post-wave 2 and wave 3. Patients with at least one of the following: hypertension, diabetes, cardiac disease, chronic pulmonary disease, asthma, chronic renal disease, any malignancy, past or active tuberculosis, HIV infection or obese, were classified as having any comorbidity. Comorbidities were either self-reported or obtained from the patient medical record.

Estimates were stratified by age group (0–14, 15–34, 35–59, 60–64, 65–69, 70–74 and 75 + years) and province. We excluded estimates based on sample sizes of <5 due to unreliability of estimates, but this generally only affected the youngest age groups as they were less likely to be hospitalized.

We estimated probabilities of ICU admission, mechanical ventilation and death in non-ICU and ICU across all wave periods based on observed data (Fig A in S1 Text). We estimated length of stay in non-ICU wards (for those never enrolled into ICU) and in ICU from admission into the ward type to recovery/discharge or death; we further estimate the time spent in non-ICU prior to ICU transfer and time in non-ICU after ICU discharge (post-ICU to recovery) (Fig A in S1 Text). We fitted a log-binomial model to the data to estimate relative risk of ICU admission and death in ICU/non-ICU, comparing "wave periods" (waves 1, 2 and 3) to "between-wave periods" (post-wave 1 and 2), while also adjusting for age, sex, comorbidities, province and public/private sector. We conducted a stratified analysis by public/private sector as the sectors differ widely in available resources and, as a result, the quality of the services they are able to provide. We tested for an interaction between wave period and public/private sector to assess for a possible effect modification of the latter. A log-binomial model was chosen as it provides unbiased estimates of the adjusted relative risk and avoids overestimation in cases where the outcomes modelled is common as is the case for logistic regression. We estimate the number of deaths that could have been avoided if the healthcare system had not been overburdened during the wave periods, by applying the age-, sector- and ward-specific between-wave probabilities of death to the number of patients admitted during a wave period. Observations with missing data were excluded from the models; the reporting of comorbidities was the variable most affected by missing data with 21% of the patient population having "unknown" comorbidity status. Due to large amounts of missing data, race was not included in the adjusted model in the main analysis, however we do a secondary analysis for the appendix in which it is included in the adjusted model. All analyses were carried out using R 4.1.2 [28].

## Results

### Study population

A total of 342,700 patients were hospitalized for COVID-19 between May 2020 and August 2021 (Figs B, C in S1 Text). Across all time periods, almost half of patients (41%-49%) admitted were aged 35–59 years, while 31%-40% of the patient population were aged 60 years or older, and 16%-24% were aged <35 years (Table 1). Overall, 55% (n = 188,461) of admitted patients were female, 51% (n = 174,916) were admitted in public sector hospitals, and 40% (n = 138,081) had at least one reported comorbidity.

### Patterns of treatment in ICU, mechanical ventilation and mortality across waves

We found patterns of higher probabilities of ICU treatment during between-wave periods, while probabilities of mortality peaked during waves and were low during between-wave periods (Fig 1, Table B in S1 Text). These patterns persisted in both public and private hospitals, and were most prominent in those aged 60+ years (Fig 1). Patterns of treatment with mechanical ventilation were less clear, but overall there was more treatment with mechanical ventilation during wave periods compared to between-wave periods (adjusted risk ratio (aRR) 1.18 [95% confidence interval (CI) 1.13–1.23]. ICU admission during a wave period was reduced by 16% compared to between-wave periods (aRR 0.84 [0.82–0.86]) (Table 2). Compared to patients aged 35–59 years, those aged 0–14 and 15–34 years were at 61% and 50% reduced risk

**Table 1. Patient characteristics stratified by time period.**

| Characteristic | | Wave 1 | Post wave 1 | Wave 2 | Post wave 2 | Wave 3 |
|---|---|---|---|---|---|---|
| | | (n = 85,743) | (n = 16,084) | (n = 107,973) | (n = 25,015) | (n = 107,885) |
| Age (years) | 0–14 | 1,904 (2%) | 724 (5%) | 2,376 (2%) | 1,172 (5%) | 3,459 (3%) |
| | 15–34 | 13,064 (15%) | 3,198 (20%) | 12,535 (12%) | 4,754 (19%) | 13,586 (13%) |
| | 35–59 | 41,907 (49%) | 7,110 (44%) | 49,582 (46%) | 10,196 (41%) | 47,630 (44%) |
| | 60–64 | 8,357 (10%) | 1,391 (9%) | 12,350 (11%) | 2,347 (9%) | 10,551 (10%) |
| | 65–69 | 6,433 (8%) | 1,122 (7%) | 10,345 (10%) | 2,092 (8%) | 9,515 (9%) |
| | 70–74 | 4,926 (6%) | 936 (6%) | 8,482 (8%) | 1,727 (7%) | 8,347 (8%) |
| | 75+ | 9,152 (11%) | 1,603 (10%) | 12,303 (11%) | 2,727 (11%) | 14,797 (14%) |
| Sex | Female | 47,767 (56%) | 8,936 (56%) | 60,267 (56%) | 14,070 (56%) | 57,421 (53%) |
| | Male | 37,895 (44%) | 7,128 (44%) | 47,635 (44%) | 10,932 (44%) | 50,407 (47%) |
| | Unknown | 81 (<1%) | 20 (<1%) | 71 (<1%) | 13 (<1%) | 57 (<1%) |
| Any comorbidity* | Yes | 38,370 (45%) | 6,163 (38%) | 45,334 (42%) | 9,945 (40%) | 38,269 (35%) |
| | No | 30,252 (35%) | 5,767 (36%) | 40,302 (37%) | 9,283 (37%) | 45,811 (42%) |
| | Unknown | 17,121 (20%) | 4,154 (26%) | 22,337 (21%) | 5,787 (23%) | 23,805 (22%) |
| Hypertension | Yes | 24,061 (29%) | 3,549 (23%) | 29,170 (28%) | 6,051 (25%) | 25,686 (25%) |
| | No | 36,516 (44%) | 7,398 (48%) | 50,011 (48%) | 11,520 (48%) | 52,819 (50%) |
| | Unknown | 22,792 (27%) | 4,599 (30%) | 25,311 (24%) | 6,544 (27%) | 26,259 (25%) |
| Diabetes | Yes | 17,865 (21%) | 2,467 (16%) | 19,956 (19%) | 3,658 (15%) | 15,267 (15%) |
| | No | 41,988 (50%) | 8,214 (53%) | 56,788 (54%) | 13,310 (55%) | 60,919 (58%) |
| | Unknown | 23,516 (28%) | 4,865 (31%) | 27,748 (27%) | 7,147 (30%) | 28,578 (27%) |
| Cardiac Disease | Yes | 2,039 (2%) | 362 (2%) | 1,603 (2%) | 377 (2%) | 1,206 (1%) |
| | No | 54,682 (66%) | 9,651 (62%) | 69,516 (67%) | 14,539 (60%) | 68,690 (66%) |
| | Unknown | 26,648 (32%) | 5,533 (36%) | 33,373 (32%) | 9,199 (38%) | 34,868 (33%) |
| Chromic Pulmonary Disease | Yes | 2,237 (3%) | 344 (2%) | 2,507 (2%) | 509 (2%) | 1,937 (2%) |
| | No | 54,162 (65%) | 9,597 (62%) | 68,242 (65%) | 14,324 (59%) | 67,717 (65%) |
| | Unknown | 26,970 (32%) | 5,605 (36%) | 33,743 (32%) | 9,282 (38%) | 35,110 (34%) |
| Asthma | Yes | 3,458 (4%) | 554 (4%) | 3,650 (3%) | 797 (3%) | 3,391 (3%) |
| | No | 54,190 (65%) | 9,530 (61%) | 68,219 (65%) | 15,032 (62%) | 68,580 (65%) |
| | Unknown | 25,721 (31%) | 5,462 (35%) | 32,623 (31%) | 8,286 (34%) | 32,793 (31%) |
| Chronic Renal Failure | Yes | 1,833 (2%) | 237 (2%) | 1,708 (2%) | 273 (1%) | 1,249 (1%) |
| | No | 54,676 (66%) | 9,723 (63%) | 68,888 (66%) | 14,483 (60%) | 68,313 (65%) |
| | Unknown | 26,860 (32%) | 5,586 (36%) | 33,896 (32%) | 9,359 (39%) | 35,202 (34%) |
| Malignancy | Yes | 575 (1%) | 107 (1%) | 383 (<1%) | 84 (<1%) | 311 (<1%) |
| | No | 55,648 (67%) | 9,807 (63%) | 70,112 (67%) | 14,671 (61%) | 69,063 (66%) |
| | Unknown | 27,146 (33%) | 5,632 (36%) | 33,997 (33%) | 9,360 (39%) | 35,390 (34%) |
| Tuberculosis | Yes | 1,081 (1%) | 310 (2%) | 990 (1%) | 387 (2%) | 664 (1%) |
| | No | 56,142 (67%) | 9,685 (62%) | 70,017 (67%) | 15,237 (63%) | 69,722 (67%) |
| | Unknown | 26,146 (31%) | 5,551 (36%) | 33,485 (32%) | 8,491 (35%) | 34,378 (33%) |
| Tuberculosis Past | Yes | 1,621 (2%) | 346 (2%) | 1,884 (2%) | 429 (2%) | 1,075 (1%) |
| | No | 54,165 (65%) | 9,402 (60%) | 67,315 (64%) | 14,074 (58%) | 66,165 (63%) |
| | Unknown | 27,583 (33%) | 5,798 (37%) | 35,293 (34%) | 9,612 (40%) | 37,524 (36%) |
| HIV Positive | Yes | 5,545 (7%) | 1,075 (7%) | 5,883 (6%) | 1,799 (7%) | 3,926 (4%) |
| | No | 52,061 (62%) | 9,009 (58%) | 65,616 (63%) | 14,123 (59%) | 67,538 (64%) |
| | Unknown | 25,763 (31%) | 5,462 (35%) | 32,993 (32%) | 8,193 (34%) | 33,300 (32%) |
| Obesity | Yes | 2,543 (3%) | 436 (3%) | 3,052 (3%) | 574 (2%) | 2,101 (2%) |
| | No | 33,331 (40%) | 1,758 (11%) | 15,519 (15%) | 3,751 (16%) | 12,258 (12%) |
| | Unknown | 47,495 (57%) | 13,352 (86%) | 85,921 (82%) | 19,790 (82%) | 90,405 (86%) |

(*Continued*)

**Table 1.** (Continued)

| Characteristic | | Wave 1 | Post wave 1 | Wave 2 | Post wave 2 | Wave 3 |
|---|---|---|---|---|---|---|
| | | (n = 85,743) | (n = 16,084) | (n = 107,973) | (n = 25,015) | (n = 107,885) |
| Sector | Private | 42,249 (49%) | 8,084 (50%) | 49,370 (46%) | 11,130 (44%) | 56,951 (53%) |
| | Public | 43,494 (51%) | 8,000 (50%) | 58,603 (54%) | 13,885 (56%) | 50,934 (47%) |
| Province | Eastern Cape | 8,878 (10%) | 2,000 (12%) | 16,317 (15%) | 1,154 (5%) | 4,437 (4%) |
| | Free State | 7,504 (9%) | 462 (3%) | 3,888 (4%) | 1,101 (4%) | 7,544 (7%) |
| | Gauteng | 25,913 (30%) | 5,861 (36%) | 23,957 (22%) | 7,681 (31%) | 48,066 (45%) |
| | KwaZulu-Natal | 13,585 (16%) | 3,359 (21%) | 24,509 (23%) | 3,882 (16%) | 9,066 (8%) |
| | Limpopo | 2,175 (3%) | 290 (2%) | 5,214 (5%) | 1,199 (5%) | 5,587 (5%) |
| | Mpumalanga | 2,420 (3%) | 951 (6%) | 4,365 (4%) | 2,659 (11%) | 4,236 (4%) |
| | North West | 6,153 (7%) | 352 (2%) | 4,561 (4%) | 2,722 (11%) | 8,846 (8%) |
| | Northern Cape | 2,425 (3%) | 142 (1%) | 1,861 (2%) | 0 (0%) | 3,103 (3%) |
| | Western Cape | 16,690 (19%) | 2,667 (17%) | 23,301 (22%) | 4,617 (18%) | 17,000 (16%) |

*Patients were classified as having a comorbidity if they had at least one of the following reported: hypertension, diabetes, cardiac disease, chronic pulmonary disease, asthma, chronic renal failure, malignancies, active tuberculosis, history of tuberculosis, HIV or obesity.

of being admitted to ICU, (aRR 0.39 [0.35–0.43] and aRR 0.50 [0.48–0.52], respectively), while those aged 60–64, 65–69 and 70–74 years were at higher risk of being admitted to ICU (aRR 1.25 [1.22–1.29], aRR 1.26 [1.23–1.30] and aRR 1.13 [1.09–1.17], respectively). Patients aged 75 years or older were less likely to be admitted into ICU (aRR 0.90 [0.87–0.93]). ICU treatment risk differed considerably by sector: Patients in public hospitals were 84% less likely (aRR 0.16 [0.16–0.17]) to be admitted to ICU compared to patients admitted in private sector hospitals, likely due to lower ICU capacity available in the public sector.

The use of mechanical ventilation changed over the course of the pandemic as treatment strategies evolved–with higher rates of use in wave 1, followed by reduced use in post wave 1, and then more consistent use over wave 2, post wave 2 and wave 3. Overall, patients were more likely to be placed on mechanical ventilation during a wave than a between-wave period (aRR 1.18 [1.13–1.23]) (Table 2). Compared to those aged 35–59, patients in the younger age groups were less likely to receive mechanical ventilation (0–14 years: aRR 0.78 [0.66–0.91], and 15–34 years: aRR 0.87 [0.82–0.93]), while patients aged 60–64 years were more likely to receive mechanical ventilation (aRR 1.05 [1.01–1.08]), and patients 75 years or older were again less likely to receive it (aRR 0.74 [0.71–0.78]).

Across all health sectors, probability of death in ICU was lowest amongst patients 0–15 years old, ranging from 0.04 to 0.08 (between-waves) and 0.09 to 0.18 (waves), and for patients aged 15–34 years ranging from 0.17 to 0.20 (between-waves) and 0.27 to 0.32 (waves) (Fig 1, Table B in S1 Text). Older age groups had higher probabilities of death in ICU; 35–59 years: 0.26–0.32 (between-waves) and 0.38–0.48 (waves), 60–64 years: 0.36–0.49 (between-waves) and 0.52–0.58 (waves), 65–69 years: 0.45–0.53 (between-waves) and 0.55–0.61 (waves), 70–74 years: 0.44–0.52 (between-waves) and 0.54–0.64 (waves), and 75+ years: 0.49–0.50 (between-waves) and 0.56–0.62 (waves). Risk of death was higher in wave periods (non-ICU: aRR 1.39 [1.35–1.43], ICU: aRR 1.31 [1.27–1.36]) compared to between-wave periods (Table 2). Overall there was an increased risk of mortality in public compared to private sector hospitals for both non-ICU and ICU admissions (aRR 2.15 [2.11–2.19] and aRR 1.10 [0.07–1.14], respectively). If the between-wave periods probability of death, specific to age group and public/private sector, was applied to hospital admissions in both non-ICU and ICU for those admitted during wave periods, the number of in-hospital deaths could possibly have been reduced by

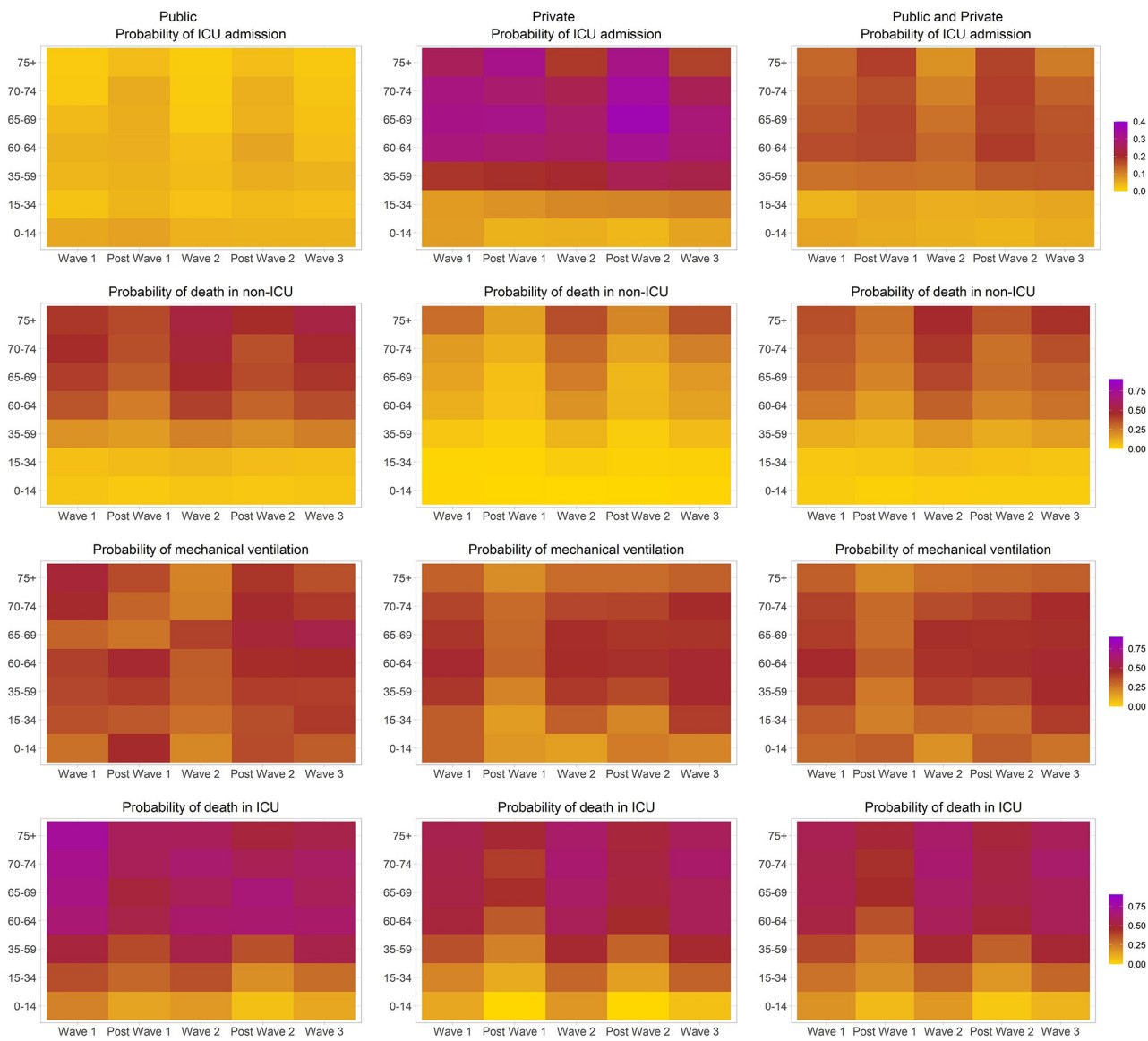

**Fig 1. Probability of ICU admission, mechanical ventilation, death in non-ICU and ICU, stratified by age, time period and public/private healthcare sector.**

approximately 19,600 [15,200–24,000] deaths, a reduction of 24% [19%-30%] between May 2020 and September 2021. Results remained similar when adjusting for race, in addition to the other variables (Table C in S1 Text). There were significant interactions between wave period and public/private sector when evaluating the outcomes ICU treatment, mechanical ventilation and death in non-ICU, but not for death in ICU (Table D in S1 Text). Interactions between province and public/private sector were mostly significant across all outcomes modelled (Table D in S1 Text). Additional analyses including individual comorbidities, in place of an aggregate measure, is included in Table E in S1 Text. Most comorbidities increased risk for ICU treatment, in particular patients with diabetes were 24% more likely to be admitted to ICU (aRR 1.24 [1.20–1.27]) (Table E in S1 Text). Similarly, the presence of most comorbidities led to increased risk of non-ICU mortality, with diabetes (aRR 1.25 [1.22–1.28]), chronic renal

**Table 2. Adjusted risk factors associated with mortality (non-ICU and ICU), treatment in ICU and mechanical ventilation.**

| Characteristic | All sectors | Public sector | Private sector | All sectors | Public sector | Private sector |
|---|---|---|---|---|---|---|
| | | *ICU treatment* | | | *Mechanical ventilation* | |
| Wave period (reference: between-wave period) | 0.84 (0.82–0.86) | 0.69 (0.63–0.75) | 0.86 (0.84–0.88) | 1.18 (1.13–1.23) | 1.01 (0.92–1.11) | 1.22 (1.17–1.28) |
| Public sector (reference: Private sector) | 0.16 (0.16–0.17) | - | - | 0.88 (0.84–0.92) | - | - |
| Male (reference: female) | 1.27 (1.25–1.29) | 1.17 (1.10–1.25) | 1.28 (1.25–1.30) | 0.99 (0.97–1.02) | 1.00 (0.93–1.06) | 0.99 (0.96–1.01) |
| Age groups (reference: 35–59 year olds) | | | | | | |
| 0–14 | 0.39 (0.35–0.43) | 1.26 (1.05–1.51) | 0.29 (0.25–0.33) | 0.78 (0.66–0.91) | 0.91 (0.76–1.08) | 0.60 (0.47–0.76) |
| 15–34 | 0.50 (0.48–0.52) | 0.72 (0.64–0.79) | 0.45 (0.43–0.48) | 0.87 (0.82–0.93) | 0.90 (0.80–1.00) | 0.85 (0.79–0.91) |
| 60–64 | 1.25 (1.22–1.29) | 0.94 (0.85–1.03) | 1.28 (1.25–1.32) | 1.05 (1.01–1.08) | 1.00 (0.91–1.11) | 1.05 (1.01–1.09) |
| 65–69 | 1.26 (1.23–1.30) | 0.66 (0.59–0.74) | 1.33 (1.29–1.37) | 0.99 (0.96–1.03) | 0.97 (0.86–1.09) | 0.98 (0.94–1.02) |
| 70–74 | 1.13 (1.09–1.17) | 0.53 (0.46–0.61) | 1.20 (1.16–1.24) | 0.96 (0.91–1.00) | 0.90 (0.77–1.05) | 0.94 (0.90–0.99) |
| 75+ | 0.90 (0.87–0.93) | 0.41 (0.36–0.47) | 0.95 (0.92–0.98) | 0.74 (0.71–0.78) | 0.84 (0.71–0.99) | 0.72 (0.68–0.75) |
| Any comorbidity (reference: no) | 1.13 (1.11–1.15) | 1.48 (1.35–1.62) | 1.11 (1.09–1.14) | 1.24 (1.21–1.27) | 0.95 (0.88–1.03) | 1.27 (1.24–1.31) |
| Provinces (reference: Western Cape) | | | | | | |
| Eastern Cape | 0.77 (0.74–0.80) | 0.30 (0.27–0.34) | 0.89 (0.85–0.93) | 1.15 (1.08–1.22) | 7.99 (6.39–10.00) | 0.89 (0.84–0.95) |
| Free State | 0.76 (0.73–0.80) | 0.53 (0.47–0.60) | 0.82 (0.78–0.86) | 1.59 (1.51–1.67) | 11.23 (9.10–13.84) | 1.25 (1.18–1.31) |
| Gauteng | 1.15 (1.12–1.18) | 0.52 (0.48–0.56) | 1.21 (1.18–1.24) | 1.13 (1.08–1.17) | 9.29 (7.56–11.43) | 0.91 (0.88–0.95) |
| KwaZulu-Natal | 0.85 (0.82–0.88) | 0.69 (0.64–0.75) | 0.88 (0.85–0.91) | 1.00 (0.96–1.05) | 6.19 (4.98–7.68) | 0.80 (0.77–0.84) |
| Limpopo | 0.59 (0.56–0.63) | 0.48 (0.42–0.56) | 0.61 (0.57–0.65) | 1.61 (1.51–1.71) | 9.23 (7.34–11.61) | 1.29 (1.20–1.38) |
| Mpumalanga | 0.82 (0.77–0.86) | 0.62 (0.54–0.71) | 0.85 (0.81–0.90) | 1.20 (1.12–1.29) | 9.93 (7.91–12.48) | 0.93 (0.86–1.00) |
| Northern Cape | 0.73 (0.68–0.79) | 0.47 (0.38–0.58) | 0.75 (0.69–0.81) | 1.90 (1.80–2.01) | 12.40 (10.00–15.38) | 1.51 (1.42–1.60) |
| North West | 0.78 (0.74–0.81) | 0.69 (0.63–0.76) | 0.77 (0.73–0.81) | 1.28 (1.20–1.36) | 5.73 (4.52–7.26) | 1.07 (1.00–1.13) |
| | | *Non-ICU mortality* | | | *ICU mortality* | |
| Wave period (reference: between-wave period) | 1.39 (1.35–1.43) | 1.27 (1.23–1.31) | 2.06 (1.93–2.21) | 1.31 (1.27–1.36) | 1.26 (1.15–1.39) | 1.32 (1.27–1.38) |
| Public sector (reference: Private sector) | 2.15 (2.11–2.19) | - | - | 1.10 (1.07–1.14) | - | - |
| Male (reference: female) | 1.14 (1.12–1.16) | 1.11 (1.09–1.13) | 1.26 (1.21–1.28) | 1.03 (1.01–1.06) | 1.01 (0.95–1.07) | 1.04 (1.02–1.06) |
| Age groups (reference: 35–59 year olds) | | | | | | |
| 0–14 | 0.17 (0.14–0.20) | 0.23 (0.19–0.27) | 0.04 (0.02–0.07) | 0.31 (0.24–0.41) | 0.42 (0.30–0.60) | 0.23 (0.15–0.34) |
| 15–34 | 0.37 (0.35–0.39) | 0.38 (0.36–0.40) | 0.28 (0.25–0.31) | 0.67 (0.62–0.72) | 0.67 (0.58–0.77) | 0.66 (0.60–0.71) |
| 60–64 | 1.75 (1.71–1.80) | 1.58 (1.53–1.62) | 2.09 (1.98–2.21) | 1.24 (1.20–1.28) | 1.20 (1.11–1.30) | 1.25 (1.21–1.29) |
| 65–69 | 2.07 (2.02–2.12) | 1.82 (1.77–1.87) | 2.65 (2.51–2.80) | 1.29 (1.25–1.33) | 1.17 (1.06–1.29) | 1.31 (1.27–1.35) |
| 70–74 | 2.30 (2.25–2.36) | 1.95 (1.90–2.01) | 3.50 (3.32–3.69) | 1.35 (1.31–1.40) | 1.18 (1.05–1.32) | 1.38 (1.34–1.43) |
| 75+ | 2.74 (2.69–2.80) | 2.16 (2.11–2.22) | 5.27 (5.07–5.47) | 1.35 (1.31–1.39) | 1.19 (1.06–1.32) | 1.37 (1.33–1.42) |
| Any comorbidity (reference: no) | 1.22 (1.20–1.25) | 1.30 (1.27–1.33) | 1.13 (1.10–1.17) | 1.08 (1.06–1.10) | 1.13 (1.03–1.24) | 1.07 (1.05–1.09) |
| Province (reference: Western Cape) | | | | | | |
| Eastern Cape | 1.35 (1.32–1.38) | 1.36 (1.32–1.40) | 1.80 (1.69–1.91) | 1.40 (1.34–1.47) | 1.14 (1.00–1.29) | 1.47 (1.40–1.54) |
| Free State | 1.01 (0.98–1.05) | 1.03 (0.99–1.07) | 1.11 (1.03–1.20) | 1.32 (1.26–1.39) | 1.15 (1.01–1.31) | 1.38 (1.31–1.45) |
| Gauteng | 1.19 (1.16–1.22) | 1.29 (1.25–1.33) | 1.13 (1.08–1.19) | 1.10 (1.06–1.14) | 1.08 (0.98–1.20) | 1.13 (1.09–1.17) |
| KwaZulu-Natal | 1.25 (1.22–1.28) | 1.27 (1.24–1.31) | 1.42 (1.34–1.50) | 1.30 (1.25–1.35) | 1.32 (1.21–1.43) | 1.32 (1.27–1.38) |
| Limpopo | 1.40 (1.36–1.44) | 1.36 (1.31–1.41) | 2.20 (1.84–2.08) | 1.49 (1.42–1.57) | 1.16 (1.00–1.34) | 1.62 (1.53–1.71) |
| Mpumalanga | 1.37 (1.32–1.42) | 1.48 (1.42–1.54) | 1.33 (1.22–1.45) | 1.28 (1.21–1.35) | 1.10 (0.94–1.30) | 1.34 (1.26–1.42) |
| Northern Cape | 1.12 (1.06–1.18) | 1.13 (1.06–1.20) | 1.24 (1.11–1.38) | 1.22 (1.13–1.32) | 0.93 (0.73–1.17) | 1.31 (1.21–1.42) |
| North West | 0.92 (0.88–0.96) | 0.93 (0.88–0.97) | 0.97 (0.89–1.05) | 1.23 (1.16–1.29) | 1.10 (0.98–1.23) | 1.29 (1.22–1.36) |

failure (aRR 1.20 [1.15–1.26]), malignancy (aRR 1.29 [1.16–1.43]), tuberculosis (aRR 1.21 [1.10–1.31]) and HIV (aRR 1.26 [1.21–1.31]) being particularly strong risk factors; with a similar pattern of comorbidities being associated with an increased risk of ICU mortality (Table E in S1 Text).

In an age-stratified analysis, patients across all age groups were less likely to be admitted to ICU during a wave period, compared to between-wave periods, with those within age groups <60 years and 60+ years having had a 9% to 12% and 19% to 37% decreased likelihood of ICU admission, respectively (Table F in S1 Text). There was a 34%-47% increased risk of non-ICU mortality in waves, compared to between-wave periods, that was similar across age groups 35–59 years (aRR 1.44 [1.36–1.53]), 60–64 years (aRR 1.47 [1.34–1.60]), 65–69 years (aRR 1.34 [1.24–1.45]), 70–74 years (aRR 1.38 [1.28–1.50]) and 75+ years (aRR 1.39 [1.31–1.47]). To a lesser extent, those aged 15–34 years were at increased risk of non-ICU death during a wave (aRR 1.15 [1.00–1.31]), compared to a between-wave period. ICU mortality risk was higher in waves, compared to between-wave periods, across all age groups; however, risk of death in ICU decreased as age increased: aRR 1.73 [1.35–2.22] for 15–34 years, aRR 1.53 [1.43–1.63] for 35–59 years, aRR 1.28 [1.17–1.41] for 60–64 years, aRR 1.16 [1.06–1.27] for 65–69 years, aRR 1.22 [1.10–1.35] for 70–74 years and aRR 1.18 [1.09–1.27] for 75+ years. This is likely reflective of the decreased risk of ICU treatment of older age groups in waves versus between-wave periods. Treatment with mechanical ventilation was more likely to occur in wave periods compared to between wave periods across all age groups, but even more so within patients aged 60 years or younger, compared to the older age groups.

## Patterns of hospital lengths of stay in non-ICU and ICU across waves

Lengths of stay remained relatively constant across the different time periods within each age group (Fig 2, Table G in S1 Text and Fig D in S1 Text for 75th percentiles). Median time to recovery from non-ICU ranged from 7 to 10 days for those aged 35+ years, and 3–8 days for those <35 years. Median time to death in non-ICU ranged from 4 to 5 days across age groups from wave 1 onwards. Time in ICU ranged from 6 to 10 days for patients who recovered, and from 7 to 11 days for patients who died. Time in non-ICU prior to ICU admission, and time in non-ICU after ICU discharge was generally very short across all waves and age groups (Table G in S1 Text). Across all age groups, private sector had longer lengths of stay for those who died in ICU and non-ICU compared to public sector, while time to recovery in ICU and non-ICU was longer in public than private sector (Table H in S1 Text). There was some variability in lengths of stay between provinces across all metrics, but no distinguishable pattern emerged (Table I in S1 Text).

## Discussion

Our analysis found that probabilities of ICU treatment, mechanical ventilation and death not only differed by age but also over time, as the health care system was put under strain by an increased number of admissions during the waves. The decrease in ICU treatment during waves is likely due to stricter triage criteria for entry due to limited capacity in hospitals. Patients who were more likely to survive were given priority of ICU beds when space for these were limited, and this is reflected in an age-stratified analysis which showed that older patients were less likely to get into ICU in waves compared to between-wave periods. This in turn would have contributed to the higher non-ICU mortality for older age groups in waves compared to between-wave periods.

The increase in mortality in both ICU and non-ICU during wave periods could be the result of a number of factors: 1) overstretched inpatient resources, including staff and oxygen, as there was an influx of patients 2) hospital admission requirements could have become stricter, allowing for only the most severe patients to be admitted, compared to the between-wave periods, 3) COVID-19 cases themselves delaying seeking treatment due to reports of hospitals being overwhelmed until symptoms became more severe. Lower use of mechanical ventilation

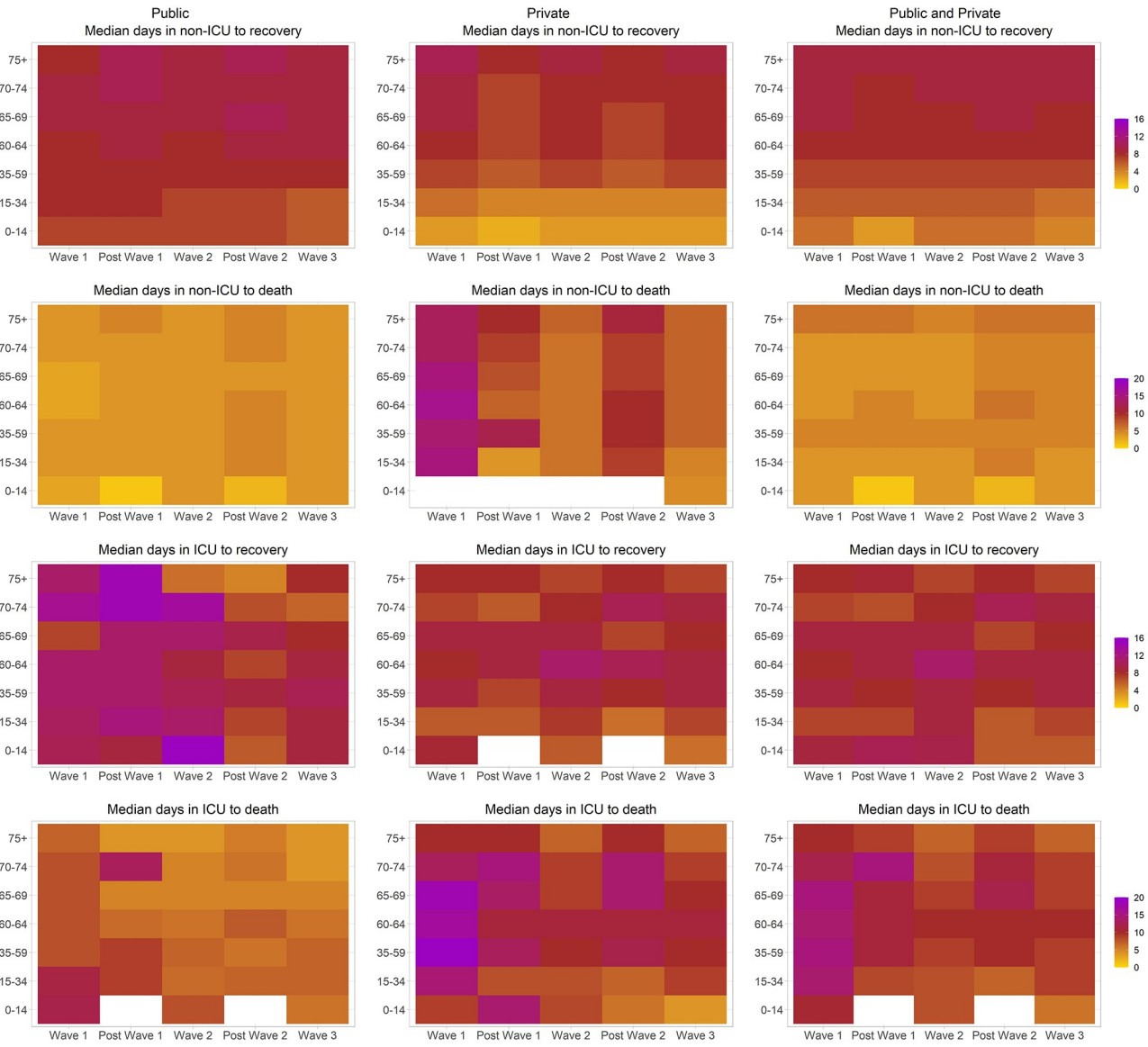

**Fig 2. Length of hospital stay (median days) in non-ICU and ICU to recovery or death, stratified by age, time period and public/private healthcare sector** *(white spaces indicate missing data).*

during between-wave periods are likely explained by an increase in milder patients being admitted to hospital as capacity eased and strict triage policies eased. We did not see as consistent a pattern of mechanical ventilation use with wave and between-wave periods as with mortality, and this may likely be due to a combination of increased ventilation capacity as more ventilators were acquired as well as a change in the use of ventilation as new therapeutics were introduced later on in the pandemic.

Our results with respect to increased mortality during periods of high community transmission and the resulting strain on the healthcare systems are similar to what has been found in high income countries such as the United States and Israel [29–31]. Irrespective of the driving factors behind the patterns of mortality during wave periods, the implications for modelling is

to include both out-of-hospital and in-hospital deaths, as in the case of the NCEM which accounts for estimated excess deaths due to COVID-19, as well as the variation of mortality over time [12].

Though the pattern distinguishing wave and between-wave time periods was consistent in both public and private sectors, patients admitted to public hospitals experienced significantly increased rates of mortality and less treatment in ICU [17]. Overall, the private sector is over-represented in the data, comprising approximately 50% of our study population, while only an estimated 15% of the South African population have private medical aid which affords them access to private healthcare [32]. Therefore, overall estimates of probabilities of treatment in ICU, mechanical ventilation and mortality need to be approached with care when applying it to other settings, and the most applicable sector needs to be considered. In addition, access to treatment is often modified by socioeconomic status (SES), with those with lower SES often having less access to hospital care than those with higher SES. This could explain the underrepresentation of public sector patients in this data. Multiple factors come into play when we consider how including SES would have affected our key parameters: lower SES could have worsened outcomes due to poorer health, higher rates of comorbidities, or an increased burden on the healthcare system that is already at capacity; or it could not have had any impact as the latter effects are removed after access to care improves their outcomes.

The DATCOV did not capture the SARS CoV-2 lineage that patients were infected with, given South Africa's limited capacity for genomic sequencing, therefore it may be possible that differences in mortality between wave periods could additionally be explained by the difference lineages circulating over time. However, we posit this would not greatly affect the patterns we observed, as the post-wave period would have still largely consisted of the COVID-19 variant of the prior wave before the new variant responsible for the subsequent wave would have started circulating [4, 33].

There are some limitations to this analysis. First, though all hospitals were reporting into the DATCOV database at the time of analysis, this was not always the case, particularly in the public sector with approximately 30% of government hospitals reporting into the system around the peak of wave 1 [34]. On joining DATCOV, hospitals were required to retrospectively capture all COVID-19 related admissions; but this might not have been completed in all hospitals. Second, our data are censored at a time when the Delta wave (wave 3) had not yet subsided, and had just passed the peak in most provinces. Therefore, estimates of probability of death are possibly higher than they would have been if we had also been able to include the downturn of admissions during the end of the wave; this would also result in the probability of ICU admission being lower than expected for the Delta wave. There is also the possibility of underestimating lengths of stay and probabilities of death as data censoring may have excluded patients who would have had longer stays and likely poorer outcomes. However, we found this was a small fraction as those still remaining in hospital in the last week of data censoring represented only 1% of all patients admitted during the Delta wave. Third, comorbidity data is self-reported for some patients, especially in public sector hospitals and/or for specific diseases. This may lead to us having either overestimated or underestimated co-prevalence. A quantitative bias analysis accounting for this self-report bias is however, beyond the scope of this paper. Fourth, we did not have data on SES; SES will influence risk of mortality, and though private/public sector might be a proxy, including SES may provide a more nuanced results and identify the most vulnerable patients within the health sector.

It is important for infectious disease modelling to consider context-specific healthcare capacity constraints, particularly in resource-limited settings, to prevent the underestimation of predicted deaths. These adjustments need to be cognisant of changes within the healthcare system that are made to either increase staff capacity or beds as the pandemic unfolds, though

this may not always be possible to know if data is limited. The prevalence of severe COVID-19, and the resulting impact on the healthcare sector, will continue changing into the future, as the vaccination roll-out programme and subsequent waves of COVID-19 increases population-level immunity, further complicated by possible waning of immunity. Future analysis will need to be conducted to describe these effects.

## Conclusion

Widespread and representative surveillance of hospital admissions and their outcomes is critical to accurate parameterization of infectious disease models. This work can, along with modelling projections and scenarios, support the work of policy makers and COVID-19 hospital readiness or management teams in understanding the implications of constrained hospital capacity on in-hospital deaths, and allow them to, within limits, boost available capacity and put in place agile processes to upscale and downscale services within pandemic waves. Further, the South African government and governments from other LMIC should engage in multisectoral activities to strengthen its health systems as part of the next steps towards pandemic preparedness.

## Supporting information

**S1 Text.**
(DOCX)

## Author Contributions

**Conceptualization:** Lise Jamieson, Harry Moultrie, Waasila Jassat.

**Data curation:** Lise Jamieson, Lucille Blumberg, Cheryl Cohen, Waasila Jassat.

**Formal analysis:** Lise Jamieson, Cari Van Schalkwyk.

**Methodology:** Lise Jamieson.

**Project administration:** Harry Moultrie.

**Supervision:** Brooke E. Nichols, Gesine Meyer-Rath.

**Validation:** Gesine Meyer-Rath, Sheetal Silal, Juliet Pulliam, Cheryl Cohen, Waasila Jassat.

**Writing – original draft:** Lise Jamieson.

**Writing – review & editing:** Lise Jamieson, Cari Van Schalkwyk, Brooke E. Nichols, Gesine Meyer-Rath, Sheetal Silal, Juliet Pulliam, Lucille Blumberg, Cheryl Cohen, Harry Moultrie, Waasila Jassat.

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
