## [Decision Letter · Decision Letter 0]

10 Jan 2023

PGPH-D-22-01374

Differential in-hospital mortality and intensive care treatment over time: Informing hospital pathways for modelling COVID-19 in South Africa

Dear Dr. Gesine Meyer-Rath,

Thank you for submitting your manuscript to PLOS Global Public Health. After careful consideration, we feel that it has merit but does not fully meet PLOS Global Public Health’s publication criteria as it currently stands. Therefore, we invite you to submit a revised version of the manuscript that addresses the points raised during the review process.

We look forward to receiving your revised manuscript.

Kind regards,

Godfrey N. Musuka, DVM, MPhil, MSc(Med), PhD

Academic Editor

Journal Requirements:

1. We have noticed that you have uploaded Supporting Information files, but you have not included a list of legends. Please add a full list of legends for your Supporting Information files after the references list. 

2. Please amend your Data Availability Statement and indicate where the data may be found.

Additional Editor Comments (if provided):

I recommend the following additional minor edits to the manuscript

1. The authors should include a separate conclusion section to their manuscript where they outline key recommendations to policy makers in their country and the region as a whole.

2. The authors have not done enough to outline key weaknesses and limitations to their study and approach.

3. The paper will benefit by having a schematic which visual shows the methodology and what the analysis is trying to achieve

4. Some key covid vaccination citations for the country are missing and one would have expected to see them included

Reviewers' comments:

Reviewer's Responses to Questions

**Comments to the Author**

1. Does this manuscript meet PLOS Global Public Health’s publication criteria? Is the manuscript technically sound, and do the data support the conclusions? The manuscript must describe methodologically and ethically rigorous research with conclusions that are appropriately drawn based on the data presented.

Reviewer #1: Yes

Reviewer #2: Yes

2. Has the statistical analysis been performed appropriately and rigorously?

Reviewer #1: Yes

Reviewer #2: Yes

3. Have the authors made all data underlying the findings in their manuscript fully available (please refer to the Data Availability Statement at the start of the manuscript PDF file)?

Reviewer #1: No

Reviewer #2: Yes

4. Is the manuscript presented in an intelligible fashion and written in standard English?

Reviewer #1: Yes

Reviewer #2: Yes

5. Review Comments to the Author

Reviewer #1: Many thanks to the authors for their clearly presented and well-written manuscript. I have some minor comments and suggestions that I have detailed below, which I hope will help improve the overall quality of this paper.

I have only a few relatively major comments (which are still quite minor), just requesting additional clarity on what has been done in parts or further information to supplement what is presented:

1. Please clarify in the Methods regarding the source of the probabilities in the Results (paragraphs starting L172 and 188) — are these estimated from the log-binomial model also, or just calculated directly from the observed data? If estimated, please include CI’s. This makes it slightly unclear which analysis is being referred to by “In an age-stratified analysis…”. Please also note that you switch from 0≤p≤1 to 0≤p≤100% between the two paragraphs.

2. On Lines 182–187, you describe the possible reduction in deaths had the probability of death in non-ICU and ICU been the same during waves as it was between waves. It would be helpful to describe a little further/more precisely how this was calculated in the methods, e.g., is it a single, baseline estimate that is then modified by the RR associated with the different waves/variants to obtain different probabilities for each period, or similar? Noting that you exclude some of your population during the Delta outbreak at the end of your study period from the analysis, it is possible that your estimates of the probability of death during this period are not accurate, and it would be helpful to expand on this in the Discussion.

3. Could you please define in the Methods whether your “ICU length of stay” is the time spent in ICU, or the total hospital length of stay for individuals who spent time in the ICU at some point during their hospital stay. On a related note, is it possible with your data to include finer resolution on the length of stay and pathways through the hospital — e.g., can you also include estimates of the median (IQR) length of stay for individuals who were admitted to non-ICU beds before progressing to ICU beds? Similarly, are there data on length of stay post ICU, in a non-ICU bed, before discharge? These kind of estimates are helpful for clinical forecasting of hospital occupancy.

4. I also would suggest that, in addition to Figure 2, you include a figure that shows a greater percentile (e.g., 75% or 90% – noting that 75% is presented in Table S3 as the upper IQR value), to give some visual indication of the variability in length of stay? This could be included in as a supplemental figure.

Minor comments and typographical errors:

Abstract, Methods: The percentages and aRR’s are swapped for death in non-ICU and ICU.

L66: Missing space between “2020which”.

L68: Capitalise ‘delta’.

L71: To provide context to the 320,000 individuals that had been vaccinated, could you please provide the population of South Africa (or perhaps size of 12+ population), here or earlier in the Introduction.

L102: Could you please clarify what exactly is meant by “the response of healthcare professionals”? Is this referring to the changing standard-of-care throughout this period?

L106: Missing “during” between “estimated the”.

L107–108: Is “…in mortality and treatment to ICU care…” meant to say “…in mortality and progression to ICU care…”?

L112: Missing the word “in” at the end of this line (i.e., “…hospitalisations in South Africa…”).

L117-119: You describe your data processing such that you allowed sufficient time for patients to experience an outcome. Could you please describe how many individuals this removed from your analysis with this processing step? I presume it is a very small fraction, but this would likely result in an underestimate of some of your outcomes (especially LoS) for those that had longer stays and potentially then worse outcomes, as related to the Delta wave.

L152–154: The statement about mechanical ventilation trends here is far less strong than the summary statement provided in the abstract. Perhaps the abstract requires slightly more caveats on this result? It would also be helpful to present the estimates (and CIs) in text as are done for other aRR’s.

L162: On the statement “…, likely due to lower ICU capacity available in the public sector”. Are there data on the number of ICU beds by sector that you could couple with the percentage of the population that have private health access, to support this statement? Appreciate that these may not be easy numbers to obtain and so not feasible.

L189: “during a waves period” should be “during a wave period”.

L197: “risk of death in ICU death decreased” should be “risk of death in ICU decreased”.

Paragraph starting line 205: spaces are missing between “to” and number throughout, e.g., “7 to10 days”.

L212: The patterns in Fig 1 suggest that the probability of mechanical ventilation increased over time (on average), across age groups. While it is very plausible just due to changing standard-of-care over time, is it possible that there were also fewer ventilators (and appropriately trained staff to operate them) available during the earlier stages of the pandemic? Is there information on ventilators purchased by the government during 2020?

Finally, I appreciate that these data may be challenging to access and so this may not be feasible, but is there any information available on hospitals reaching capacity during any of the waves that could support statements about ICU admissions, death etc. changing as capacity are reached? E.g., just a number/proportion of hospitals that reached their “effective bed” (i.e., beds and staff) capacity during each wave?

Reviewer #2: Thank you for giving me the opportunity to review this important manuscript. I appreciate the author's effort in doing this research. I have a list of suggestions/comments, which are listed below.

Introduction:

1. In paragraph 3, the authors discuss current research in low- and middle-income nations. Though there is limited research in these countries, it would be helpful for readers if the author could search for more recent research on low- and middle-income countries and include it here.

2. It would also be useful to briefly describe research conducted in middle- and high-income countries, as this would help readers understand what is known and where gaps exist from a global perspective.

Methods:

1. A data extraction flowchart would be helpful for readers.

2. “Patients with at least one of the following: hypertension, diabetes, cardiac disease, chronic pulmonary disease, asthma, chronic renal disease, any malignancy, past or active tuberculosis, HIV infection or obese, were classified as having any comorbidity.” Self-reported? Self-reporting of diagnoses is a common way to collect health data, and patient-reported comorbidities are reliable, especially for chronic conditions (e.g., diabetes). Depending on the disease and other factors, patients' reports may underestimate or overestimate relevant comorbidities. Was an attempt made to account for bias in this study? This would be helpful to include in this paragraph.

3. A binary variable called comorbidity has been created and used in this study. Have the authors investigated whether patients with a single condition (e.g., diabetes or asthma) had a higher risk of ICU admission or mortality? In addition to the comorbidity variable, I would recommend performing analysis with a specific condition in the model because not all conditions have the same degree of influence on ICU admission and mortality.

4. In Table 1, please include statistics on specific conditions (hypertension, diabetes, cardiac disease, chronic pulmonary disease, asthma, chronic renal disease, any malignancy, past or active tuberculosis, HIV infection, or obesity).

5. It would be beneficial to provide justification for the log-binomial model. Despite its benefits, there are disadvantages (e.g., boundedness of parameter space). The reader would benefit from a brief explanation of why log-binomial models were chosen over other models (such as logistic regression) for this study.

6. The model description paragraph needs to include information about model development steps. This study took two steps. This study first considered model outcome and wave period while adjusting for age, sex, comorbidities, province, and public/private sector. This study considered stratified analysis by public/private sector in the second step.

7. Before conducting stratified analysis, has this study investigated the interaction between "wave period" and "public/private sector"?

8. Has this research examined the relationship between "public/private sector" and "province"? It is essential because the number of hospitals can vary by province.

9. This model building paragraph should also contain information about what was done with the "Unknown" category of the comorbidity variable during development of model.

Results:

1. Having trend graphs for all of the most important outcome variables will help visualise the trend of ICU admission and mortality. On the graph, it would also be useful to highlight COVID waves and restrictions period.

2. The result section can be improved further by including information about the additional analysis suggested in the methods section's comments.

3. The paragraph "Patterns of hospital lengths of stay in non-ICU and ICU across waves" may include statistics on how length of stay differed by hospital type (i.e., public versus private) and provinces.

Discussion:

1. In paragraph 4, there is a discussion about the type of hospital. This paragraph can be improved by including a discussion of hospital accessibility and its potential impact on the outcome. Patients from different socioeconomic backgrounds are likely to have varying levels of access to hospitals, which may influence ICU admission and mortality.

2. The limitation section also needs to highlight the limitations of using self-reported comorbidity and not having data on socioeconomic status at the individual level and residential area level. Socioeconomic status is likely to influence various health conditions, hospital service access, and utilisation. I understand that these data may not be available but acknowledging them would be beneficial for future research as well as interpretation of current research findings.

3. Policy recommendations can be strengthened further by providing information on multisectoral approaches to strengthening health systems.

6. PLOS authors have the option to publish the peer review history of their article (what does this mean?). If published, this will include your full peer review and any attached files.

**Do you want your identity to be public for this peer review?** For information about this choice, including consent withdrawal, please see our Privacy Policy.

Reviewer #1: No

Reviewer #2: No

---

## [Editor Report · Decision Letter 1]

16 Feb 2023

Differential in-hospital mortality and intensive care treatment over time: Informing hospital pathways for modelling COVID-19 in South Africa

PGPH-D-22-01374R1

Dear Gesine Meyer-Rath,

We are pleased to inform you that your manuscript 'Differential in-hospital mortality and intensive care treatment over time: Informing hospital pathways for modelling COVID-19 in South Africa' has been provisionally accepted for publication in PLOS Global Public Health.

Best regards,

Godfrey N Musuka, DVM, MPhil, MSc(Med), PhD

Academic Editor